# Scarlet Flax *Linum grandiflorum* (L.) In Vitro Cultures as a New Source of Antioxidant and Anti-Inflammatory Lignans

**DOI:** 10.3390/molecules26154511

**Published:** 2021-07-27

**Authors:** Bushra Asad, Taimoor Khan, Faiza Zareen Gul, Muhammad Asad Ullah, Samantha Drouet, Sara Mikac, Laurine Garros, Manon Ferrier, Shankhamala Bose, Thibaut Munsch, Duangjai Tungmunnithum, Arnaud Lanoue, Nathalie Giglioli-Guivarc’h, Christophe Hano, Bilal Haider Abbasi

**Affiliations:** 1Department of Biotechnology, Quaid-i-Azam University, Islamabad 45320, Pakistan; bushra.mirza16@gmail.com (B.A.); mrtaimoor39@gmail.com (T.K.); faizazareengul@gmail.com (F.Z.G.); asad_ullah8050@yahoo.com (M.A.U.); 2Laboratoire de Biologie des Ligneux et des Grandes Cultures (LBLGC), INRA USC1328, Université d’Orléans, CEDEX2, 45067 Orléans, France; samantha.drouet@univ-orleans.fr (S.D.); sara.mikac93@gmail.com (S.M.); laurine.garros@univ-orleans.fr (L.G.); duangjai.tun@mahidol.ac.th (D.T.); 3EA2106 Biomolécules et Biotechnologies Végétales, Université de Tours, 37200 Tours, France; manon.ferrier@univ-tours.fr (M.F.); shankhamala85@gmail.com (S.B.); thibaut.munsch@idorsia.com (T.M.); arnaud.lanoue@univ-tours.fr (A.L.); nathalie.guivarch@univ-tours.fr (N.G.-G.); 4Department of Pharmaceutical Botany, Faculty of Pharmacy, Mahidol University, Bangkok 10400, Thailand

**Keywords:** *Linum grandiflorum*, plant growth regulators, callus culture, plant specialized metabolites, lignans, neolignans, antioxidant, cyclooxygenase inhibitors, anti-inflammatory

## Abstract

In vitro cultures of scarlet flax (*Linum grandiflorum* L.), an important ornamental flax, have been established as a new possible valuable resource of lignans and neolignans for antioxidant and anti-inflammatory applications. The callogenic potential at different concentrations of α-naphthalene acetic acid (NAA) and thidiazuron (TDZ), alone or in combinations, was evaluated using both *L. grandiflorum* hypocotyl and cotyledon explants. A higher callus induction frequency was observed on NAA than TDZ, especially for hypocotyl explants, with a maximum frequency (i.e., 95.2%) on 1.0 mg/L of NAA. The presence of NAA (1.0 mg/L) in conjunction with TDZ tended to increase the frequency of callogenesis relative to TDZ alone, but never reached the values observed with NAA alone, thereby indicating the lack of synergy between these two plant growth regulators (PGRs). Similarly, in terms of biomass, NAA was more effective than TDZ, with a maximum accumulation of biomass registered for medium supplemented with 1.0 mg/L of NAA using hypocotyls as initial explants (DW: 13.1 g). However, for biomass, a synergy between the two PGRs was observed, particularly for cotyledon-derived explants and for the lowest concentrations of TDZ. The influence of these two PGRs on callogenesis and biomass is discussed. The HPLC analysis confirmed the presence of lignans (secoisolariciresinol (SECO) and lariciresinol (LARI) and neolignan (dehydrodiconiferyl alcohol [DCA]) naturally accumulated in their glycoside forms. Furthermore, the antioxidant activities performed for both hypocotyl- and cotyledon-derived cultures were also found maximal (DPPH: 89.5%, FRAP 866: µM TEAC, ABTS: 456 µM TEAC) in hypocotyl-derived callus cultures as compared with callus obtained from cotyledon explants. Moreover, the anti-inflammatory activities revealed high inhibition (COX-1: 47.4% and COX-2: 51.1%) for extract of hypocotyl-derived callus cultures at 2.5 mg/L TDZ. The anti-inflammatory action against COX-1 and COX-2 was supported by the IC_50_ values. This report provides a viable approach for enhanced biomass accumulation and efficient production of (neo)lignans in *L. grandiflorum* callus cultures.

## 1. Introduction

Plants have been widely used as a reservoir of key phytochemicals having a broad range of medicinal and cosmetic purposes throughout human history [1,2,3,4]. *L. grandiflorum,* commonly known as Scarlet flax due to its brightly colored flowers, belongs to the *Linaceae* family [5]. It is native to Algeria but can be found in North Africa and Southern Europe as indigenous flora, besides, it has been introduced in many other parts of the world (GRIN-USDA), especially because it is now cultivated as an ornamental species. Health benefits, including antiproliferative action against cancer cells and anti-inflammatory activities, have been reported for extracts from this plant [5,6,7,8] but only a few reports focused on its phytochemical potential [8,9]. *Linum* species are known as one of the lucrative sources of valuable and diverse anticancer, antioxidant and anti-inflammatory lignans [10,11,12,13,14].

Plant-specialized metabolites that are responsible for health attributes have high demand in pharmaceutical industries [3,15,16]. But most of the time, the production capacity of the natural source does not meet industrial criteria for direct exploitation. Low accumulation levels and contents variability from natural habitat and inappropriate extraction methods of these metabolites make their development very challenging [3,17]. Alternatively, plant tissue culture technology can be commercially exploited for its capacity to enhance metabolite productions. Competitive benefits of these cultures include rapid production of phytochemicals, irrespective of environmental and seasonal constraints, without geographical limits, disease free, easy harvesting and specific material production make plant tissue culturing highly desirable [18]. Research has emphasized developing strategies for improved growth of plants with sufficient yield of medicinal compounds to reduce the threats of depleting plant resources, raised by the overexploitation of plants in their natural habitat [15]. Various plant growth regulators (PGRs) have been explored and found with altering growth, morphology, and metabolite accumulation in callus cultures [19]. Similar reports of collection and extraction of essential phytochemicals in several industrially important species, including *Panax ginseng*, *Taxus spp.*, *Fagonia indica*, *Eclipta alba* and *Silybum marianum,* were reported globally [20,21,22,23,24]. Several *Linum* species have also been previously exploited for their tremendous biosynthetic potential in in vitro platform [12,13,16,25,26,27]. However, no reports are available on establishment of feasible in vitro cultures and production phytochemicals by *L. grandiflorum*.

The theory of free radical aging is based on the observation that reactive oxygen and nitrogen species (ROS/RNS) can cause oxidative damage, cell malfunction, and physiological decline, eventually leading to aging, degenerative diseases and death [28]. Although mechanisms for mending oxidatively damaged macromolecules exist in human cells, some damage remains. During their growth and/or exposure to stress, plants produce a variety of active phytochemicals, including lignans, which act as natural antioxidants [4]. The antioxidant actions of compounds are mainly attributed to their redox characteristics, which allow them to behave as reducing agents or hydrogen atom donors [29]. Inflammation also plays a role in the development of degenerative diseases, and anti-inflammatory phytochemicals are commonly found in plant extracts [30,31,32,33]. Their ability to block important enzymes involved in the inflammation process, such as cyclooxygenase-1 and cyclooxygenase-2 (COX-1 and COX-2), determines their anti-inflammatory capability. COXs, in particular, are key players and targets in the inflammation process for the development of nonsteroidal anti-inflammatory medicines. Prostaglandin E2 is produced by COX-2, the endogenous pain causing molecule. However, COXs play a multifaceted role in platelet and renal homeostasis, as well as gastrointestinal tissue homeostasis, and their deregulation has been related with the onset of certain cancers [34]. Consequently, there is an active search for new drugs that selectively block the COX-2 enzyme and with little side effects. Several lignans and related compounds have been described as possible anti-inflammatory compounds by targeting some of these key enzymes in vitro [30,31,32,33].

The current study was aimed to develop a competent protocol for establishing in vitro cultures of *L. grandiflorum* on MS media using different PGRs applied at different concentrations. In addition to assess PGRs’ callogenesic effects on two different explant sources (i.e., hypocotyls and cotyledons), quantification of the main (neo)lignans have been performed by HPLC. Only a few reports dealing with *L. grandiflorum* phytochemical analysis from seeds and leaves are available in the literature [5,8,9]. Particularly for *Linum usitatissimum* (L.) and other related species from the genus *Linum*, Schmidt et al. [9] reported the presence of the secoisolariciresinol (SECO) under its diglucoside form (secoisolariciresinol diglucoside, SDG) in the seeds of *L. grandiflorum*. However, the present study is the very first report on HPLC-based quantification of (neo)lignans in both hypocotyl- and cotyledon-derived callus cultures grown under various PGRs concentrations. Moreover, to explore the biological activities of extracts deriving from these different in vitro cultures, their antioxidant potential, but also their anti-inflammatory activity, determined by their inhibition capacity toward both cyclooxygenase 1 and 2 (COX1 vs. COX2), were evaluated. This study provides the first step toward the development of a new potent bioproduction system for multifunctional health-promoting bioactive (neo)lignans.

## 2. Results

### 2.1. Callus Induction and Morphogenesis

Cotyledon and hypocotyl explants of scarlet flax (*L. grandiflorum* cv. Rubrum) were placed onto MS medium supplemented with several concentrations of PGRs alone or in combination for evaluation of callus induction frequency. Callus formation was noticed for both explant types for almost all the concentrations applied (Figure 1) with variable callus induction response (Appendix A).

Highest callus induction frequencies were recorded for hypocotyl-and cotyledon-derived callus cultures in presence of 1.0 mg/l NAA (95% and 87%, respectively). Contrarily, TDZ applications resulted in lower induction frequencies for callogenesis used either alone or combined with NAA in both type of explants. In agreement with this result, several reports observed that TDZ alone prohibited callus formation [24,35]. This could be linked with stress induced by higher levels and/or suppression of endogenous hormones [36]. The MS medium devoid of any PGR or elicitor could not trigger callus formation in any explant. These results are in agreement with available reports [15,27]. Previous studies supported the considerable influence of PGRs concentrations toward callogenesis frequency in other *Linum* species [37,38]. The callus induction of NAA, either alone or in combination with TDZ, was previously reported [39,40,41]. Our results revealed that, for scarlet flax, hypocotyl explants were more prone to callogenesis than cotyledon explants. This behavior has already been observed for other *Linum* species [27,42].

Both hypocotyl- and cotyledon-derived callus cultures established at different concentrations of PGRs were investigated for morphological variations. For hypocotyl-derived callus under NAA treatments, the calli were found friable in texture and dark green in appearance, whereas in cotyledon explants, more compact callus was seen on NAA with less moisture content and dark green complexion. Contrarily, the cultured TDZs, induced in vitro and obtained from both explant types, were found slightly to moderately compact, with low moisture content and were yellowish to greenish in appearance (Figure 1). Similar observations were noted by Anjum et al. [43] and Ullah et al. [35] for in vitro cultures of other *Linum* species. The results are also in correspondence with explant-based variations reported for other medicinal plants, such as *Corydalis saxicola* [44].

### 2.2. Biomass Accumulation

The callus cultures initiated from the hypocotyl and cotyledon explants were investigated for biomass accumulation. The different culture conditions resulted in various levels of biomass accumulation as a function of the PGRs treatments (Figure 2).

The biomass accumulation was maximal in hypocotyl-derived callus cultures. In case of hypocotyl explants, callus cultures were found more responsive with the maximum biomass accumulation in presence of 1.0 mg/L NAA (DW: 13 g/L), while the minimal biomass value was recorded at 10 mg/L TDZ + 1.0 mg/L NAA in combination (DW: 2.76 g/L). The cotyledon-derived callus cultures comparatively showed optimum biomass accumulation with 1.0 mg/L NAA treatment (DW: 11.4 g/L). But lower biomass production was observed for cotyledon-derived callus than hypocotyl-derived callus obtained under the same conditions (Figure 2). For NAA, above 1 mg/L, a continuous decrease in biomass accumulation was observed. On the other hand, biomass production significantly increased with increasing concentrations of TDZ when applied alone, whereas addition of NAA annihilated this trend. A possible explanation for this pattern could rely on the suppression of endogenous biosynthesis and/or signaling of endogenous hormones/signals in the explant material, or different responses of tissues can also trigger this response of explants to different growth stimulators [36,45]. In absence of PGRs, no effective biomass production was noticed for both hypocotyl- and cotyledon-derived callus cultures of *L. grandiflorum.*

The observed differential biomass accumulation among cultures under various PGRs could be due to physiological and biochemical potency of explant type and tissues. Many factors behind callus proliferation include explant type, plant genotype, growth conditions and concentration of PGRs applied [17,45]. Our results are supported by previous studies that applied different concentrations of NAA alone or combined with other PGRs for biomass production of in vitro cultures [41,46,47,48].

### 2.3. Total Phenolics and (Neo)lignans Accumulations

Both total phenolic content (TPC) and HPLC quantification of (neo)lignans for scarlet flax callus under each PGRs condition were determined (Appendix A; Table 1).

For callus deriving from hypocotyl explants, the highest TPC (4.7 mg/g DW) was recorded in callus grown on 1.0 mg/L NAA, while the least accumulation (2.2 mg/g DW) occurred on 10 mg/L TDZ (Appendix A). Likewise, for cotyledon-derived callus, the optimum TPC (4.3 mg/g DW) was recorded for 1.0 mg/L NAA, whereas the minimal production (1.1 mg/g DW) was recorded for 10 mg/L TDZ. Complex response was observed when combined PGRs were used, thus supporting the preference of employing PGRs alone as previously reported [40,49,50,51]. Interestingly, here, maximum TPC can be related with the optimal biomass production (Appendix A). This relation found in biomass and phytochemicals accumulation was reported for other medicinal plant species [37,38,49,52].

This is the very first report on HPLC-based quantification of (neo)lignans in both hypocotyl- and cotyledon-derived callus cultures grown under various PGRs concentrations (Table 1). Not surprisingly, considering its phylogenic classification among *Linum* species [53], HPLC analysis revealed the presence of the 8-8′ lignans secoisolariciresinol (SECO) and lariciresinol (LARI), and the 8-5′ neolignan dehydrodiconiferyl alcohol (DCA) (Appendix A). Maximum SECO contents were recorded respectively in hypocotyl- (5.3 mg/g DW) and cotyledon- (2.6 mg/g DW) derived callus cultures at 2.5 gm/L TDZ, while maximum LARI accumulations were recorded for both type of explant cultures at 1.0 mg/L NAA (4.4 mg/g DW and 3.2 mg/g DW, respectively). The DCA accumulation levels were maximum (67.6 mg/g DW) for hypocotyl- and (32.7 mg/g DW) for cotyledon-derived callus at PGR concentration 2.5 g/L TDZ.

Only a few studies have investigated the phytochemical composition of *L. grandiflorum* seeds and leaves [5,8,9]. Schmidt et al. [9] solely reported the presence of SECO under its diglucoside form (SDG) in the seeds of *L. grandiflorum*. The significant accumulation of this lignan is compatible with the classification of *L. grandiflorum* within Linaceae from the genus *Linum* [9]. However, the current report is pioneer on HPLC-based quantification of SECO in both hypocotyl- and cotyledon-derived callus cultures grown under various PGRs concentrations. In addition, another lignan (LARI) and neolignan (DCA) were also detected and quantified for the first time in *L. grandiflorum*. Interestingly, the accumulation of large quantities of these two (neo)lignans has already been observed in several in vitro cultures of *L. usitatissimum*, another Linaceae from the *Linum* section, despite the fact that they are accumulated at a much lower level than SECO in naturally grown plants [2,12,25,26,27,38,54,55,56,57,58,59,60,61,62,63,64]. However, the present study is the very first report on HPLC-based quantification of (neo)lignans in both hypocotyl- and cotyledon-derived callus cultures grown under various PGRs concentrations. Efficient production of (neo)lignans under NAA and/or TDZ treatments has been previously observed for other *Linum* species [12,25,54]. Here, overall, the (neo)lignan productions were found enhanced, in correlation with biomass accumulation, especially on NAA treatment, with a significant correlation observed for LARI (Figure 2; Appendix A). In agreement with this observation, (neo)lignans have been previously reported to stimulate plant growth and/or cell division [65].

### 2.4. Antioxidant and Anti-Inflammatory Activities

Three different assays, including DPPH, FRAP and ABTS, were performed to measure the antioxidant activities of extracts deriving from the cultures of *L. grandiflorum* maintained with different PGRs concentrations (Table 2).

The highest DPPH free radical scavenging activity (FRSA) values were recorded for extracts from hypocotyl-derived callus grown in presence of NAA with maximum activity (90.5% FRSA) for 1.0 mg/L NAA. Contrarily, highest FRAP and ABTS antioxidant activities were recorded for extracts from hypocotyl-derived callus cultures grown in presence of 2.5 mg/L TDZ. A similar trend was observed for cotyledon-derived callus, but with lower antioxidant potential as compared to hypocotyl-derived callus grown under the same conditions. This is the first report on the investigation of the antioxidant potential of *L. grandiflorum* callus cultures. Under both developmental and stress conditions, the plant cells produce reactive oxygen species (ROS). Produced under uncontrolled levels, these ROS can induce damages on DNA and other biomolecules and threat cell viability [66]. To cope with these ROS, plants produce a wide array of antioxidant phytochemicals [66]. Lignans are well-known antioxidants [67], and in vitro plant culture is recognized as a rich source of these lignan-antioxidants [22,38]. Here, to confirm this trend, significant correlations connected the *L. grandiflorum* (neo)lignans with the antioxidant assays, with highest correlations obtained for LARI with DPPH assay, SECO for ABTS assay and DCA for FRAP assays (Table 3). These correlations could reveal distinct antioxidant mechanisms for these (neo)lignans, since it is established that ABTS assay reveals antioxidants acting through the hydrogen atom transfer mechanism (HAT), FRAP assay reveals an electron transfer mechanism (ET), whereas mixed mechanism (HAT and ET) is revealed by DPPH assay [29]. The antioxidant potential of these compounds has been previously evaluated [67,68,69].

The anti-inflammatory potential of extracts deriving from the cultures of *L. grandiflorum*, were evaluated against COX-1 and COX-2 activities (Table 4).

Highest inhibitions were recorded for extract deriving from hypocotyl-derived callus grown on 2.5 mg/L TDZ for both COX-1 (47.4%) and COX-2 (51.1%). Comparatively, the standard drug Ibuprofen (10 µM), used as positive controls, resulted in enzymatic activity inhibition of 31.4 ± 0.8% and 29.8 ± 1.2% of COX-1 and COX-2, respectively. Inflammation occurs as immune response to pathogens, harmful stimuli, irritates and damaged cells [70]. COXs have been extensively used to study anti-inflammatory potentials of plant extracts [71]. Plants produced a wide array of phytochemicals with anti-inflammatory potentials [72,73,74]. These phytochemicals in plants were confirmed accountable for enzymatic inhibition that triggers inflammation in vivo [72]. Here, (neo)lignans, accumulated in *L. grandiflorum* extracts, may be responsible for the COXs inhibition. The highest significant correlations were observed for SECO with COX-1 inhibition, and DCA with COX-2 inhibition (Table 4). However, given the high concentration of DCA in *L. grandiflorum* callus extracts compared to SECO and LARI, it is worth emphasizing that this neolignan is most likely to be responsible for the COXs (especially COX2) inhibitions.

To the best of our knowledge, the individual inhibition capacity toward COX-1 vs. COX-2 of these (neo)lignans has never been studied. To confirm our correlation study, using purified (neo)lignans, the IC_50_ against COX-1 and COX-2 for each compound was determined (Table 5; Figure 3).

Optimum inhibition of COXs capacities were recorded for SECO toward COX-1 (IC_50_ = 21.7 µM) and for DCA toward COX-2 (19.2 µM), thus confirming the anti-inflammatory potential of *L. grandiflorum* extracts and our correlation analysis (Table 3). Interestingly, the specificity toward COX-1 vs. COX-2 was clearly different for lignans vs. neolignans (Table 3) and shows that these compounds could be attractive scaffolds for the development of specific COXs inhibitors. However, the low selectivity of COX-1/COX-2 ratio would cause some gastrointestinal side effects [75]. Therefore, the development and application of SECO and LARI should be considered carefully. The greater specificity of DCA for inhibiting COX-2 might be of more interest.

A principal component analysis (PCA) was performed as unsupervised analysis to summarize relevant changes of (neo)lignans accumulations, biomass accumulation and biological activities according to explant origin and PGRs (Figure 4).

The PCA score of the first two components explained 77.1% of the variation (Figure 4A) with the first principal component (PC1) accounting for 61.1% and the second (PC2) for 16%. PCA showed no clear discrimination according to the origin of explant. The loading plot (Figure 4B) showed the projection of the variables on the two first components. Clearly, (neo)lignans accumulations (LDG, DCG and SDG), antioxidant (DPPH, ABTS and FRAP) and anti-inflammatory (COX-1 and COX-2) activities were projected together on PC1 positive, whereas biomass accumulation was projected on PC2 positive. PCA confirmed in a rapid outlook the efficiency of NAA alone to stimulate biomass accumulation, and that high biological activities of the plant extracts were associated to high (neo)lignans accumulations.

## 3. Materials and Methods

### 3.1. Chemicals

All the extraction solvents were analytical and supplied by Sigma-Aldrich (Saint-Quentin Fallavier, France). Lignans and neolignan standards were prepared as described previously [12]. All other reagents were purchased from Sigma-Aldrich (Saint-Quentin Fallavier, France).

### 3.2. Seed Germination

*Linum grandiflorum* (cv Rubrum) commercial seeds (Vilmorin & Cie) were used. Seed sterilization and germination were performed following previously established protocol for flax seeds [2] with little modifications. Initially, seeds were washed to remove any dust on the seed surface, followed by selection of viable seeds via the free floating technique. The seeds were then sterilized with (1.0% *w/v*) mercuric chloride for 40 s, followed by dipping in (70% *w/v*) ethanol for 1 min. The seeds were then washed with sterile double-distilled water three times and placed on sterilized filter paper. The sterilized seeds were placed on solid Murashige and Skoog (MS) medium [76] containing agar (8 g/L), sucrose (30 g/L) and pH adjusted to 5.7. Seeds were then placed in a growth room at 25 ± 2 °C temperature, using light intensity of 40 μmol/m^2^/s at around a 16/8 h (light/dark) photoperiod.

### 3.3. Establishment of Callus Cultures

Both the cotyledon (0.5 cm) and hypocotyl (1.0 cm) explants of *L. grandiflorum* of 5-day old seedlings, grown in vitro, were employed as the source of the explants and inoculated on three different experimental media containing sucrose (30g/L), agar (8 g/L) along with various concentrations of α-naphthaleneacetic acid (NAA: 0.1, 1.0, 2.5, 5.0, 10 mg/L), thidiazuron (TDZ: 0.1, 1.0, 2.5, 5.0, 10 mg/L) and a combination of both (TDZ: 0.1, 1.0, 2.5, 5.0, 10 mg/L and NAA: 0.1 mg/L constant) and placed in a growth room at 25 ± 2 °C temperature, using light intensity of 40 μmol/m^2^/s at around a 16/8 h (light/dark) photoperiod. Four explants were inoculated for each condition. All experiments were conducted twice for each treatment. Media without PGRs addition were used as controls. The cultures were harvested at day 30 after inoculation. Biomass determination (fresh weights (FW) and dried weight (DW)) were determined for each condition. For dry weight (DW) estimation, cells were frozen and lyophilized 48 h.

### 3.4. Sample Extraction

The plant extract was prepared following the protocol of [12] with slight modifications. Briefly, 50 mg of lyophilized powdered callus was mixed with 500µL of MeOH, followed sonication for 20 min at 45 °C and an ultrasonic frequency of 45 kHz in an ultrasonic bath (Prolabo). The sample was vortexed for 5 min and the overall procedure was repeated twice for efficient extraction of phytochemicals. Then, the sample was centrifuged at 10,000 rpm for 15 min and the supernatant was evaporated to dryness in a SpeedVac concentrator (Thermo Fisher) at 40 °C. The resulting pellet was resuspended in 500 µL of 0.1 M citrate-phosphate pH 4.8 buffer containing 5 unit/mL *β*-glucosidase from almonds (Sigma) for lignan and neolignan aglycone release for 4 h at 37 °C. The sample was finally centrifuged at 10,000 rpm for 15 min and the supernatant filtered (0.45 µm) prior to further analyses (phytochemical analysis and biological assays). Note that an aliquot of each extract was also analyzed by HPLC, without enzymatic treatment, to appreciate the relative proportion of aglycones vs. glycosides. The extracts were at −80 °C.

### 3.5. Phytochemical Assays for Estimation of Secondary Metabolites

#### 3.5.1. High-Performance Liquid Chromatography (HPLC) Analysis

The lignans (secoisolariciresinol (SECO) and lariciresinol (LARI) and neolignans (dehydrodiconiferylic alcohol [DCA]) contents in *L. grandiflorum* extracts were determined by HPLC as described in Anjum et al. [12], using a Varian HPLC system composed of Prostar 230 pump, Metachem Degasit, Prostar 410 autosampler and Prostar 335 Photodiode Array Detector (PAD) driven by Galaxie version 1.9.3.2 software. Separation of compounds was carried out using a Purospher (Merck), RP-18 column (250 × 4.0 mm i.d. 5 µm) at 40 °C. Validation of this separation method, including calibration curves, LOD, LOQ and R^2^, are described in Anjum et al. [12].

#### 3.5.2. Total Phenolic Contents

To ascertain total phenolic contents, Folin–Ciocalteu’s reagent was used (absorbance was measured at 630 nm with UV-visible spectrophotometer), as described in Anjum et al. [38].

### 3.6. In Vitro Antioxidant Assays

#### 3.6.1. Free Radical Scavenging Activity

Antioxidant potential of callus extracts was determined by following the protocol reported in Anjum et al. [38]. The reaction mixture was prepared in microplates by adding 20 μL of each callus extract with DPPH solution at a quantity of 180 μL, plus DPPH solution at a quantity of 180 μL. The plate was then incubated in the dark for an hour, and absorbance measured at 517 nm on a microplate reader. To calculate FRSA, the following formula was applied: % scavenging DPPH free radical = 100 × (1 − AE/AD), with AE = absorbance of the mixture at 517 nm and AD = absorbance of the DPPH only.

#### 3.6.2. Ferric-Reducing Antioxidant Power Assay (FRAP)

The protocol described in detail by Nazir et al. [29] was used for the assessment of reducing the power of extracts. Briefly, the FRAP solution (190 μL) was first prepared (10 mM TPTZ, 20 mM FeCl_3_, 6H_2_O and 300 mM acetate buffer pH 3.6; ratio 1:1:10 (*v/v/v*)). The reaction mixture was prepared by adding 10 μL of the plant extract to 190 µL of FRAP solution. The mixture was incubated at (25 ± 1 °C) for 15 min. A BioTek Synergy II absorbance microplate reader was used for measurement of absorption at 630 nm. The assay was performed in triplicate and reducing potential expressed in Trolox C-equivalent antioxidant activity (TEAC).

#### 3.6.3. ABTS Antioxidant Assay

ABTS antioxidant activity was performed according to the procedure described in detail by Nazir et al. [77]. In brief, the mixture was prepared by dissolving ABTS salt (7 mM) in potassium persulfate (2.45 mM). Then, the solution was placed for 16 h in the dark, and after that the absorbance (734 nm) was adjusted to 0.7 prior to its use. The plant extract (10 µL) was added to 190 µL ABTS solution after measuring its absorbance at 734 nm. The incubation period, under complete dark, was 15 min. Absorbance was determined with a BioTek Instruments (Synergy II) microplate reader at 734 nm. This test was carried out in triplicate and antioxidant potential was expressed in TAEC.

### 3.7. Anti-Inflammatory COX-1 and COX-2 Inhibition Activities

COX-1 (Ovine) and COX-2 (Human) inhibition assays were performed using the corresponding Cayman Chem kits. The assays were performed according to the manufacturer’s instructions. Arachidonic acid (1.1 mM) was used as substrate. Reaction was followed at 590 nm in a BioTek Instrument Synergy II microplate reader. Ibuprofen (10 µM) was used as a commercial inhibitor as a control. SECO, LARI and DCA were prepared as described previously [12] and their respective IC_50_ values against COX1 and COX2 were determined using ED50plus v1.0 software.

### 3.8. Statistical Analysis

All the experiments were independently performed in, at least, triplicate under the same environmental conditions. Data is expressed as mean ± SE of three independent replicates. Significant differences between groups were determined by ANOVA, followed by two-tailed multiple *t*-tests with Bonferroni correction, performed with XL-STAT 2019 biostatistics software (Addinsoft). All results were considered significant at *p* < 0.05, represented by different letters. Principal component analysis (PCA) was performed using SIMCA P+ version 15.0 (Umetrics AB, Umeå, Sweden). Variables were mean-centered and unit variance-scaled prior to PCA. Hierarchical clustering analysis and Pearson correlation coefficient analysis were obtained with PAST 3.0, with significant thresholds at *p* < 0.05, *p* < 0.01 and *p* < 0.001 represented by *, ** and ***, respectively.

## 4. Conclusions

The current study aimed to develop a protocol to exploit *L. grandiflorum* in vitro cultures for the bioproduction of essential secondary metabolites, using hypocotyl and cotyledon explants. Various concentrations of PGRs either alone or in combination were employed, using both type of explants. Among PGRs, the NAA alone was found very efficient in callus induction and stimulated biomass accumulation, along with total phenolic and (neo)lignans contents, anti-inflammatory (COX-1 and COX-2) and antioxidants activities (DPPH, FRAP and ABTS). Overall, the hypocotyl explants at 1.0 mg/L NAA allowed optimal biomass levels, as compared with cotyledon explants against other PGRs, either alone or in combination. HPLC analysis showed optimum production levels of lignans (SECO and LARI) and neolignan (DCA) in callus culture of *L. grandiflorum.* These (neo)lignans can constitute attractive scaffolds for the future design of specific inhibitors of COX-1 vs. COX-2, as supported by their IC_50_ against both enzymes. Hence, a competent protocol was established that could help contribute to upcoming research domains using cell suspension cultures of *L. grandiflorum*. For instance, this can be utilized as an efficient production system for medicinal mass production needed for future phytopharmaceutical industries.

## Figures and Tables

**Figure 1 molecules-26-04511-f001:**
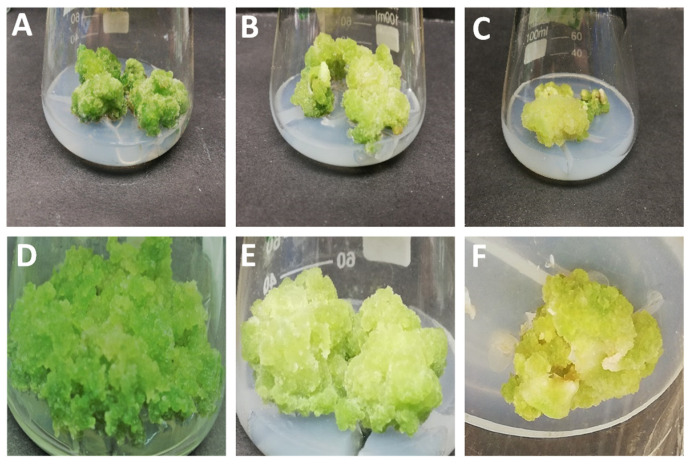
Morphological aspects of callus cultures of *L. grandiflorum* from stem explants (**A**) 1.0 mg/L NAA (**B**) 1.0 mg/L NAA + 2.5 mg/L TDZ (**C**) 2.5 mg/L TDZ, and from leaf explants (**D**) 1.0 mg/L NAA, (**E**) 1.0 mg/L NAA + 2.5 mg/L TDZ (**F**) 2.5 mg/L TDZ.

**Figure 2 molecules-26-04511-f002:**
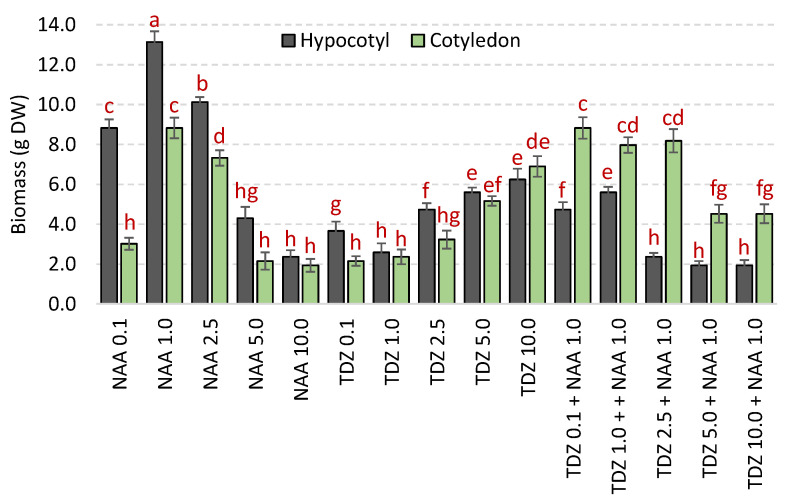
Comparison of biomass dry weight (DW) accumulation for hypocotyl- and cotyledon-derived callus cultures of *L. grandiflorum* grown on various plant growth regulators (PGRs; i.e., NAA and/or TDZ) treatments. NAA and TDZ concentrations are indicated in mg/L. Different letters indicate significant difference (*p* < 0.05).

**Figure 3 molecules-26-04511-f003:**
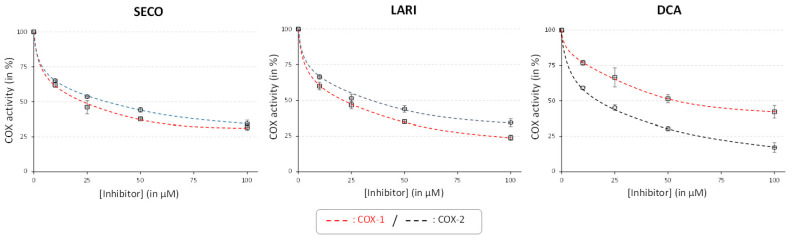
IC_50_ curves for the inhibition COXs by the lignans (SECO and LARI) and neolignan (DCA) accumulated in *L. grandiflorum* in vitro cultures.

**Figure 4 molecules-26-04511-f004:**
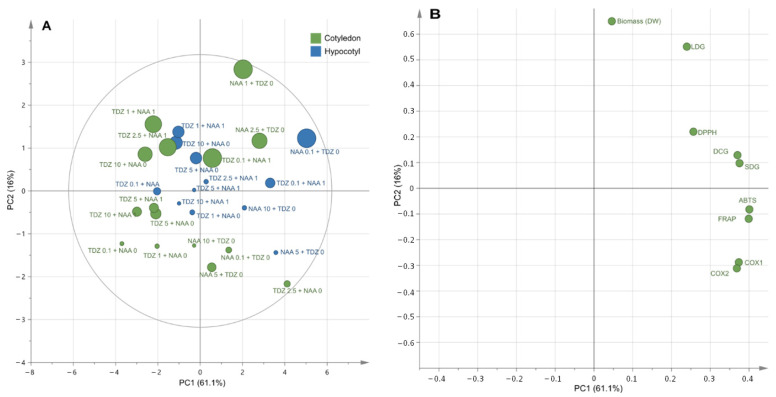
Principal component analysis (PCA) for the discrimination of the different *L. grandiflorum* extracts derived from callus cultures as a function of their phytochemical compositions and biological activities, with circle size relative to biomass, expressed as dry weight. Score plot (**A**); loading plot (**B**). Explained variance by factor 1 (PC1) = 61.1% and by factor 2 (PC2) = 16%.

**Table 1 molecules-26-04511-t001:** Lignans and neolignans accumulation in in vitro cultures of *L. grandiflorum* (L.) as a function of PRG concentrations.

PRGs (mg/L)	SECO		LARI		DCA	
	Hypocotyl	Cotyledon	Hypocotyl	Cotyledon	Hypocotyl	Cotyledon
NAA 0.1	3.4 ± 0.2 ^bc^	1.7 ± 0.2 ^d^	2.5 ± 0.2 ^c^	1.2 ± 0.1 ^ef^	43.9 ± 3.6 ^bc^	22.7 ± 2.4 ^ef^
NAA 1.0	3.7 ± 0.4 ^b^	1.8 ± 0.3 ^de^	**4.4 ± 0.3** ^a^	3.2 ± 0.1 ^b^	48.0 ± 4.7 ^b^	26.8 ± 1.4 ^e^
NAA 2.5	3.5 ± 0.2 ^b^	1.9 ± 0.2 ^d^	4.1 ± 0.1 ^a^	3.1 ± 0.3 ^bc^	45.9 ± 2.1 ^b^	26.0 ± 3.4 ^e^
NAA 5.0	1.5 ± 0.3 ^de^	0.6 ± 0.3 ^f^	2.7 ± 0.2 ^c^	1.3 ± 0.2 ^ef^	16.6 ± 6.0 ^fg^	7.9 ± 3.6 ^gh^
NAA 10.0	2.5 ± 0.2 ^c^	1.3 ± 0.1 ^ef^	2.2 ± 0.3 ^cd^	1.4 ± 0.2 ^e^	31.4 ± 5.0 ^de^	16.9 ± 2.5 ^fg^
TDZ 0.1	1.1 ± 0.4 ^ef^	0.4 ± 0.1 ^f^	1.7 ± 0.1 ^de^	0.9 ± 0.1 ^f^	13.1 ± 2.1 ^g^	5.2 ± 1.9 ^h^
TDZ 1.0	1.6 ± 0.3 ^de^	0.7 ± 0.2 ^f^	2.0 ± 0.3 ^d^	1.0 ± 0.2 ^ef^	15.3 ± 2.5 ^g^	11.0 ± 0.8 ^g^
TDZ 2.5	**5.3 ± 0.1** ^a^	2.6 ± 0.3 ^c^	3.1 ± 0.2 ^b^	1.5 ± 0.3 ^de^	**67.6 ± 2.9** ^a^	32.7 ± 2.7 ^de^
TDZ 5.0	1.5 ± 0.1 ^de^	0.8 ± 0.1 ^f^	2.3 ± 0.1 ^c^	1.3 ± 0.1 ^e^	21.0 ± 1.8 ^f^	9.4 ± 1.3 ^h^
TDZ 10.0	1.2 ± 0.3 ^def^	0.6 ± 0.2 ^f^	2.6 ± 0.4 ^cd^	2.1 ± 0.2 ^d^	15.3 ± 6.7 ^fg^	7.9 ± 1.3 ^h^
TDZ 0.1 + NAA 1.0	2.8 ± 0.3 ^c^	1.5 ± 0.1 ^e^	2.4 ± 0.3 ^cd^	1.9 ± 0.1 ^d^	35.6 ± 3.9 ^cd^	16.6 ± 0.8 ^g^
TDZ 1.0 + + NAA 1.0	1.2 ± 0.2 ^def^	0.7 ± 0.2 ^ef^	2.4 ± 0.2 ^cd^	2.1 ± 0.1 ^d^	14.0 ± 3.0 ^g^	7.9 ± 1.5 ^h^
TDZ 2.5 + NAA 1.0	1.6 ± 0.1 ^de^	0.8 ± 0.3 ^ef^	2.0 ± 0.1 ^d^	1.3 ± 0.2 ^e^	26.9 ± 1.9 ^e^	9.4 ± 3.1 ^gh^
TDZ 5.0 + NAA 1.0	1.5 ± 0.1 ^de^	0.9 ± 0.2 ^f^	2.0 ± 0.1 ^d^	1.0 ± 0.2 ^ef^	19.6 ± 1.5 ^fg^	9.0 ± 1.7 ^h^
TDZ 10.0 + NAA 1.0	1.4 ± 0.2 ^de^	0.4 ± 0.3 ^f^	1.6 ± 0.2 ^de^	0.8 ± 0.2 ^f^	18.3 ± 3.6 ^fg^	7.6 ± 2.6 ^gh^

PRGs: plant growth regulators; SECO: secoisolariciresinol; LARI: lariciresinol; DCA: dehydrodiconiferyl alcohol. Values are means ± SD from three replicates expressed in mg/g DW. Highest contents for each compound are indicated in bold. Different superscript letters indicate significant difference (*p* < 0.05).

**Table 2 molecules-26-04511-t002:** Antioxidant potential of *L. grandiflorum* extracts from callus cultures (derived from hypocotyl and cotyledon explants) grown under different PGRs treatments (in mg/L).

PRGs (mg/L)	DPPH ^1^		ABTS ^2^		FRAP ^2^	
	Hypocotyl	Cotyledon	Hypocotyl	Cotyledon	Hypocotyl	Cotyledon
NAA 0.1	80.9 ± 3.2 ^bc^	73.0 ± 2.2 ^d^	326.9 ± 26.0 ^cd^	269.2 ± 17.3 ^d^	582.5 ± 40.8 ^c^	407.8 ± 33.4 ^de^
NAA 1.0	**90.5 ± 4.0** ^a^	84.1 ± 3.8 ^ab^	423.1 ± 19.2 ^b^	336.5 ± 21.2 ^cd^	737.9 ± 68.0 ^d^	504.8 ± 56.3 ^cd^
NAA 2.5	80.9 ± 1.9 ^c^	77.8 ± 2.9 ^c^	346.2 ± 13.5 ^c^	259.6 ± 18.0 ^d^	601.9 ± 31.1 ^c^	427.2 ± 42.7 ^de^
NAA 5.0	76.2 ± 3.2 ^cd^	66.7 ± 4.1 ^de^	365.4 ± 20.2 ^c^	288.5 ± 25.3 ^d^	679.6 ± 52.4 ^bc^	330.1 ± 52.4 ^e^
NAA 10.0	71.4 ± 2.4 ^d^	60.3 ± 2.4 ^e^	230.8 ± 17.3 ^de^	201.9 ± 15.2 ^e^	427.2 ± 40.8 ^d^	310.7 ± 27.2 ^e^
TDZ 0.1	52.4 ± 3.5 ^ef^	42.9 ± 1.7 ^g^	96.2 ± 19.2 ^gh^	48.1 ± 10.6 ^h^	194.2 ± 62.1 ^f^	97.1 ± 25.2 ^fg^
TDZ 1.0	55.6 ± 3.3 ^ef^	50.8 ± 2.7 ^f^	153.9 ± 22.1 ^fg^	115.4 ± 16.8 ^g^	388.4 ± 56.3 ^de^	291.3 ± 40.8 ^ef^
TDZ 2.5	69.8 ± 2.4 ^d^	60.3 ± 3.3 ^e^	**490.4 ± 18.3** ^a^	355.8 ± 24.0 ^c^	**970.9 ± 20.8** ^a^	679.6 ± 48.5 ^bc^
TDZ 5.0	54.0 ± 1.7 ^ef^	41.3 ± 1.8 ^g^	163.5 ± 10.6 ^h^	117.3 ± 16.7 ^g^	330.1 ± 21.8 ^e^	291.3 ± 25.2 ^e^
TDZ 10.0	46.0 ± 4.0 ^f^	39.7 ± 3.8 ^g^	135.6 ± 19.7 ^fg^	86.5 ± 23.2 ^gh^	271.8 ± 60.2 ^ef^	135.9 ± 33.0 ^fg^
TDZ 0.1 + NAA 1.0	69.8 ± 2.7 ^d^	58.7 ± 4.0 ^e^	269.2 ± 17.1 ^d^	214.4 ± 21.2 ^e^	582.5 ± 44.7 ^c^	427.2 ± 15.5 ^d^
TDZ 1.0 + + NAA 1.0	74.6 ± 2.1 ^cd^	63.5 ± 2.9 ^de^	115.4 ± 14.5 ^g^	94.2 ± 18.7 ^gh^	213.6 ± 34.9 ^f^	116.5 ± 42.7 ^fg^
TDZ 2.5 + NAA 1.0	85.7 ± 1.4 ^b^	76.2 ± 4.3 ^bcd^	153.9 ± 8.7 ^f^	120.2 ± 23.8 ^fg^	252.4 ± 15.5 ^e^	213.6 ± 64.1 ^ef^
TDZ 5.0 + NAA 1.0	77.8 ± 1.6 ^c^	61.9 ± 3.3 ^e^	144.2 ± 6.7 ^f^	101.9 ± 20.9 ^fg^	291.3 ± 21.4 ^e^	194.2 ± 41.2 ^fg^
TDZ 10.0 + NAA 1.0	71.4 ± 1.9 ^d^	60.3 ± 3.5 ^e^	134.6 ± 11.0 ^f^	67.3 ± 19.6 ^h^	271.8 ± 31.1 ^ef^	174.8 ± 52.4 ^fg^

^1^ DPPH is expressed in % of free radical scavenging activity; ^2^ ABTS and FRAP are expressed in µM Trolox-C equivalent antioxidant activity (TEAC). PRGs: plant growth regulators. Highest antioxidant activities for each assay are in bold. Values are means ± SD from three replicates. Different superscript letters indicate significant difference (*p* < 0.05).

**Table 3 molecules-26-04511-t003:** Pearson correlation coefficients (PCC) showing the relation between the main phytochemicals and the biological activities (antioxidant and anti-inflammatory) of extracts of in vitro cultures of *L. grandiflorum* (L.).

Biological Assay	SECO	LARI	DCA	TPC
DPPH	0.523 *	**0.646** ***	0.555 *	0.639 ***
ABTS	**0.837** ***	0.650 **	0.833 ***	0.713 ***
FRAP	0.872 ***	0.627 *	**0.890** ***	0.685 ***
COX-1	**0.670** ***	0.344 **	0.683 ^ns^	0.470 **
COX-2	0.679 **	0.352 **	**0.696** ***	0.474 **

Highest correlations are indicated in bold. Significance level: * *p* < 0.05, ** *p* < 0.01, *** *p* < 0.001, ns: not significant.

**Table 4 molecules-26-04511-t004:** Anti-inflammatory potential against COX-1 and COX-2 (in inhibition percentage) of *L. grandiflorum* extracts from callus cultures (derived from hypocotyl and cotyledon explants) grown under different PGRs treatments (in mg/L).

PRGs (mg/L)	COX-1		COX-2	
	Hypocotyl	Cotyledon	Hypocotyl	Cotyledon
NAA 0.1	31.1 ± 2.3 ^bc^	23.1 ± 2.1 ^de^	34.2 ± 2.4 ^bc^	27.2 ± 2.1 ^cd^
NAA 1.0	12.3 ± 1.1 ^fg^	10.1 ± 0.5 ^g^	14.4 ± 1.2 ^fg^	13.7 ± 1.1 ^g^
NAA 2.5	33.0 ± 2.3 ^bc^	28.1 ± 2.2 ^cd^	37.1 ± 2.4 ^b^	31.7 ± 2.2 ^c^
NAA 5.0	34.2 ± 2.3 ^bc^	30.0 ± 2.3 ^c^	38.4 ± 2.6 ^b^	32.3 ± 2.3 ^c^
NAA 10.0	21.5 ± 1.5 ^de^	18.1 ± 1.5 ^e^	24.6 ± 1.9 ^de^	20.1 ± 1.9 ^e^
TDZ 0.1	8.9 ± 0.4 ^gh^	7.6 ± 0.3 ^h^	10.8 ± 0.6 ^h^	9.8 ± 0.4 ^h^
TDZ 1.0	14.1 ± 1.2 ^f^	11.2 ± 1.1 ^g^	17.0 ± 1.5 ^ef^	14.4 ± 1.3 ^fg^
TDZ 2.5	**47.4 ± 2.8** ^a^	35.9 ± 2.5 ^b^	**51.1 ± 2.9** ^a^	39.8 ± 2.6 ^b^
TDZ 5.0	13.2 ± 1.1 ^fg^	11.0 ± 0.9 ^g^	16.7 ± 1.4 ^ef^	14.9 ± 1.4 ^fg^
TDZ 10.0	10.0 ± 0.7 ^g^	9.4 ± 0.7 ^g^	13.4 ± 1.1 ^g^	12.7 ± 1.2 ^g^
TDZ 0.1 + NAA 1.0	24.1 ± 1.6 ^d^	19.3 ± 1.8 ^e^	28.1 ± 2.1 ^cd^	22.7 ± 2.0 ^de^
TDZ 1.0 + + NAA 1.0	9.1 ± 0.3 ^g^	8.1 ± 0.3 ^h^	12.6 ± 1.1 ^g^	12.1 ± 1.1 ^gh^
TDZ 2.5 + NAA 1.0	13.3 ± 1.3 ^fg^	10.2 ± 0.9 ^g^	17.0 ± 1.4 ^ef^	14.4 ± 1.4 ^fg^
TDZ 5.0 + NAA 1.0	11.0 ± 1.1 ^g^	10.0 ± 0.5 ^g^	15.6 ± 1.2 ^f^	13.6 ± 1.2 ^fg^
TDZ 10.0 + NAA 1.0	9.9 ± 0.5 ^g^	8.3 ± 0.3 ^h^	11.2 ± 0.8 ^gh^	10.1 ± 0.8 ^gh^

Highest inhibition values for each enzyme are in bold. Values are means ± SD from three replicates. Ibuprofen (10 µM) was used as positive control for COX-1 and COX-2 activity, leading to enzyme inhibition of 31.4 ± 0.8% and 29.8 ± 1.2%, respectively. Different superscript letters indicate significant difference (*p* < 0.05).

**Table 5 molecules-26-04511-t005:** IC_50_ (µM) values for COX inhibition by lignans and neolignan accumulated in *L. grandiflorum* in vitro cultures.

Compound	COX-1 IC_50_		COX-2 IC_50_		Specificity (COX-1/COX2 IC_50_ Values)
	(in µM)	(in µg/mL)	(in µM)	(in µg/mL)	
SECO	**21.7****± 1.9** ^a^	**59.9** ** ± 5.2 ** ^a^	32.7 ± 0.8 ^b^	90.2 ± 2.2 ^b^	0.67 ± 0.01 ^b^
LARI	24.2 ± 6.9 ^a^	67.1 ± 19.1 ^a^	34.6 ± 2.9 ^b^	96.0 ± 8.0 ^b^	0.70 ± 0.04 ^b^
DCA	51.3 ± 0.3 ^b^	144.0 ± 0.8 ^b^	**19.2****± 2.5** ^a^	**53.6** ** ± 7.0 ** ^a^	2.67 ± 0.05 ^a^

Best inhibition values (i.e., lowest IC_50_ values) for each enzyme are in bold. Values are means ± SD from three replicates. Different superscript letters indicate significant difference (*p* < 0.05).

## Data Availability

All of the data supporting the findings of this study are included in this article.

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
