# Peer review of "Scarlet Flax Linum grandiflorum (L.) In Vitro Cultures as a New Source of Antioxidant and Anti-Inflammatory Lignans"

_molecules, 2021, doi:10.3390/molecules26154511_

Round 1

Reviewer 1 Report

As far as I know this is a first report of in establishment of in vitro culture of  L. grandiflorum and their use for production of phytochemicals. In general I do not have problem with experiments performed and how the paper is written. I would just like some clarification of minor points.

The language in general is not bad, but some minor language corrections are necessary. Some mistakes remain and this distracts from reading.   So I would suggest careful reading or even better additional check by native speaker or professional editing service. 

The list of abbreviations will be useful as not every reader will be familiar with short names of lignans like SECO and LARI and those are not mentioned in the introduction and if someone skipped abstract then it can be problematic. There are some editing mistakes- like sentences with a word was deleted by mistake.

I would like to see some additional information how lignans were identified. There is no mention of standards used.

It is not exactly clear if the lignans composition in this Linum species was known and if so maybe this could be added to the introduction.  Some discussion on how the extracts from callus varies from those form naturally grown plants. Is  the lignan profile the same?

Author Response

REVIEWER 1

As far as I know this is a first report of in establishment of in vitro culture of  L. grandiflorum and their use for production of phytochemicals. In general I do not have problem with experiments performed and how the paper is written. I would just like some clarification of minor points.

AUTHORS: Thank you very much for your very useful comments, we have revised our manuscript accordingly.

The language in general is not bad, but some minor language corrections are necessary. Some mistakes remain and this distracts from reading.   So I would suggest careful reading or even better additional check by native speaker or professional editing service. 

AUTHORS: The needful has been done.

The list of abbreviations will be useful as not every reader will be familiar with short names of lignans like SECO and LARI and those are not mentioned in the introduction and if someone skipped abstract then it can be problematic. There are some editing mistakes- like sentences with a word was deleted by mistake.

AUTHORS: Abbreviations are now specified not only in the Abstract, but also in the text where they first appear, as well as in the legends of Figures and Tables. We have corrected editing mistakes and a native English speaker carefully read the manuscript.

I would like to see some additional information how lignans were identified. There is no mention of standards used.

AUTHORS: Sorry we had mistakenly provided the wrong reference (same author and same year, hence our confusion) for the standards and the identification method. Standards were prepared as described previously by our group. The correct reference is: Anjum, S., Abbasi, B. H., Doussot, J., Favre-Réguillon, A., & Hano, C. (2017). Effects of photoperiod regimes and ultraviolet-C radiations on biosynthesis of industrially important lignans and neolignans in cell cultures of Linum usitatissimum L.(Flax). Journal of Photochemistry and Photobiology B: Biology, 167, 216-227. We have corrected this mistake in the revised version of our manuscript.

It is not exactly clear if the lignans composition in this Linum species was known and if so maybe this could be added to the introduction.  Some discussion on how the extracts from callus varies from those form naturally grown plants. Is  the lignan profile the same?

AUTHORS: Thank you for the interesting remarks. We have included this information in the Introduction and discussed it in the Results and Discussion section. To date only few reports deal with L. grandiflorum phytochemical analysis from seeds and leaves (Mohammed, et al., 2009a; Mohammed, et al., 2009b; Mohammed, et al., 2010; Schmidt et al. 2010). In particular, as for Linum usitatissimum (L.) and other related linaceae from the Linum section, Schmidt et al (2010) reported the presence of the secoisolairciresinol (SECO) under its diglucoside form (secoisolairciresinol diglucoside, SDG) in the seeds of L. grandiflorum. However, the present study is the very first report on HPLC based quantification of (neo)lignans in both hypocotyl- and cotyledon-derived callus cultures grown under various PGRs concentrations.

Reviewer 2 Report

The manuscript entitled “ Scarlet flax Linnum grandiflorum (L.) in vitro cultures as a new source of antioxidant and anti-inflammatory lignans” is suitable for the publication in Molecules after the major revision.

The results are interesting, the manuscript is well written,  but there are few things to improve before acceptation

  • The main objection - Fig 3 - dependence of the degree of inhibition of cyclooxygenase (1 and 2) in% inhibition on the concentration of pure lignans (in micromoles) - please explain why the higher the concentration, the lower the% inhibition - is it usually not the other way round ???
  •  
  • 2 – biomass accumulation and lines – 153-154 – authors wrote that “minimal biomass was recorded at 10 mg/L TDZ + 1.0 mg/L NAA in combination (DW: 2.76 g/L)” – the sample TDZ 5.0 + NAA 1.0 does not differ statistically from the previous one - anyway, there are a few more hypocotyl samples that have a similar value (all marked with the letter h) - this should be included in the description of the results
  • 2 - the unit of concentration of PGRs should be given either in the figure caption or under the x axis
  • the abbreviated names of the compounds should be explained under table 1
  • a very high concentration of DCA was marked compared to SECO and LARI - it is worth emphasizing that it is the neolignans that inhibit COX2 more effectively
  • it would be worth adding HPLC charts in the supplement - at least for the best PRG, besides the values given in the table in the text differ (line 198) - and it is not about rounding to 1 place decimal point in the table
  • table 3 – please clarify - in table 3 - Pearson's correlations between COX-1 and DCA were marked as insignificant, why?
  • has the correlation between antioxidant and anti-inflammatory properties and the content of polyphenols and flavonoids been checked? Maybe these properties result not only from the presence of lignans?
  • in Table 1, the concentration of lignans is expressed in mg/g - while pure compounds in micromoles - is it possible to establish any relationship - explain how are these values in relation to each other? On what basis were the concentrations of pure lignans selected?

Author Response

REVIEWER 2

The manuscript entitled “ Scarlet flax Linnum grandiflorum (L.) in vitro cultures as a new source of antioxidant and anti-inflammatory lignans” is suitable for the publication in Molecules after the major revision.

The results are interesting, the manuscript is well written,  but there are few things to improve before acceptation

AUTHORS: Thank you very much for your very useful comments, we have revised our manuscript accordingly.

  • The main objection - Fig 3 - dependence of the degree of inhibition of cyclooxygenase (1 and 2) in% inhibition on the concentration of pure lignans (in micromoles) - please explain why the higher the concentration, the lower the% inhibition - is it usually not the other way round ???

AUTHORS: The y axis represent COX activity not inhibition. We have corrected this mistake in the revised version of the Figure 3.

  • 2 – biomass accumulation and lines – 153-154 – authors wrote that “minimal biomass was recorded at 10 mg/L TDZ + 1.0 mg/L NAA in combination (DW: 2.76 g/L)” – the sample TDZ 5.0 + NAA 1.0 does not differ statistically from the previous one - anyway, there are a few more hypocotyl samples that have a similar value (all marked with the letter h) - this should be included in the description of the results

AUTHORS: Thank you, we agree with your remark. However, we would like to provide in the text the higher and the lowest values recorded. Therefore, we have revised the text accordingly.

  • 2 - the unit of concentration of PGRs should be given either in the figure caption or under the x axis

AUTHORS: Thank you for this remark, we have added the unit of PRGs concentration (mg/L) in the figure legend.

  • the abbreviated names of the compounds should be explained under table 1

AUTHORS: The meaning of each abbreviation is now provided in the legend of this table.

  • a very high concentration of DCA was marked compared to SECO and LARI - it is worth emphasizing that it is the neolignans that inhibit COX2 more effectively

AUTHORS: We have integrated this important remark in the revised version of the manuscript. [“However, given the high concentration of DCA in L. grandiflorum callus extracts compared to SECO and LARI, it is worth emphasizing that this neolignan is most likely to be responsible for the COXs (especially COX2) inhibitions.”]

  • it would be worth adding HPLC charts in the supplement - at least for the best PRG, besides the values given in the table in the text differ (line 198) - and it is not about rounding to 1 place decimal point in the table

AUTHORS: A typical HPLC chromatogram is now provided as Figure S2 (supplementary materials) and the concentrations values have been corrected. Instead of mean values, values from the first replicate were provided. We apologize for the error.

  • table 3 – please clarify - in table 3 - Pearson's correlations between COX-1 and DCA were marked as insignificant, why?

AUTHORS: The p-value is 0.0564, therefore the correlation between DCA and COX-1 inhibition is not considered as statistically significant in the present experimental conditions.

  • has the correlation between antioxidant and anti-inflammatory properties and the content of polyphenols and flavonoids been checked? Maybe these properties result not only from the presence of lignans?

AUTHORS: The correlation value is now provided in revised Table 3.

  • in Table 1, the concentration of lignans is expressed in mg/g - while pure compounds in micromoles - is it possible to establish any relationship - explain how are these values in relation to each other? On what basis were the concentrations of pure lignans selected?

AUTHORS: The concentration of lignans for IC50 are now also in µg/mL as well in order to ease comparison. The concentration range is the most commonly utilized range in pharmacological studies. It is unusual to exceed 100 M to evaluate the inhibition of a pure compound; otherwise, this typically indicates that the molecule under consideration is inactive on the target examined.

Reviewer 3 Report

The manuscript uncovered the enhanced biomass accumulation strategy of lignanoid glucosides from in vitro cultures of scarlet flax (Linum grandiflorum L.). This study provided a remarkable in vitro and environment-friendly source of bioactive lignanoid glucosides. Although the results would be of some interest, several indistinct sentences remained to be cleared.

First of all, the evaluated compounds [lines 34 and 35: secoisolariresinol (SECO) and lariciresinol (LARI) and dehydrodiconiferyl alcohol (DCA) in their glucoside forms] were ambiguous. SECO, LARI and DAC are aglycones. Author mentions about “in their glucoside forms” means? There could be many kinds of glucoside forms of SECO, LARI, and DCA. In your reference 38, the article showed precise compounds as SDG (Secoisolariciresinol diglucoside, LDG (Lariciresinol diglucoside)…etc. Therefore, the compound’s purchase source also should be mentioned in the manuscript to give more accuracy to the standards.

Second, Figure 2 could be more detailed and clear if the author adds the explanations for each superscript (a, b, bc, de, ef….). As a result, readers will have no ideas about their meanings.

About the anti-inflammatory activity, SECO, LARI, and DCA showed inhibition toward COX-1 and COX-2. However, there’s no positive control to compare for the inhibitory effect. Furthermore, the low selectivity of COX-1/COX-2 ratio would cause some gastrointestinal side effects1. Therefore, the development and application should be considered carefully.

Besides the above suggestion, there are many format mistakes in the reference part. The scientific name should be italic (references 2, 5, 8, 12, 13, 16, 17, 19-21, 24, 26, 27, 29, 30, 32, 33, 35-52, 63-65). Reference 13 lacks the journal name. The absence of volume or page should be noticed (references 32, 33, 64, 67).

Reference

  1. Baigent C, Patrono C. Selective cyclooxygenase 2 inhibitors, aspirin, and cardiovascular disease: A reappraisal. Arthritis & Rheumatism 2003; 48: 12-20

Author Response

REVIEWER 3

The manuscript uncovered the enhanced biomass accumulation strategy of lignanoid glucosides from in vitro cultures of scarlet flax (Linum grandiflorum L.). This study provided a remarkable in vitro and environment-friendly source of bioactive lignanoid glucosides. Although the results would be of some interest, several indistinct sentences remained to be cleared.

AUTHORS: Thank you very much for your very useful comments, we have revised our manuscript accordingly.

First of all, the evaluated compounds [lines 34 and 35: secoisolariresinol (SECO) and lariciresinol (LARI) and dehydrodiconiferyl alcohol (DCA) in their glucoside forms] were ambiguous. SECO, LARI and DAC are aglycones. Author mentions about “in their glucoside forms” means? There could be many kinds of glucoside forms of SECO, LARI, and DCA. In your reference 38, the article showed precise compounds as SDG (Secoisolariciresinol diglucoside, LDG (Lariciresinol diglucoside)…etc. Therefore, the compound’s purchase source also should be mentioned in the manuscript to give more accuracy to the standards.

AUTHORS: Sorry we had mistakenly provided the wrong reference (same author and same year, hence our confusion) for the standards preparation, extraction and the identification method. Standards were prepared as described previously by our group. The correct reference is: Anjum, S., Abbasi, B. H., Doussot, J., Favre-Réguillon, A., & Hano, C. (2017). Effects of photoperiod regimes and ultraviolet-C radiations on biosynthesis of industrially important lignans and neolignans in cell cultures of Linum usitatissimum L. (Flax). Journal of Photochemistry and Photobiology B: Biology, 167, 216-227. We have corrected this mistake in the revised version of our manuscript. Because lignans can accumulate in various glycoside forms, (neo)lignan aglycones were released after β-glucosidase (from almonds, Sigma-Aldrich) to simplify the chromatograms. Because aglycones were not detected in the absence of enzymatic digestion, we concluded that (neo)lignans are initially stored as glycosides.

Second, Figure 2 could be more detailed and clearer if the author adds the explanations for each superscript (a, b, bc, de, ef….). As a result, readers will have no ideas about their meanings.

AUTHORS: Different letters indicate significant difference (p < 0.05). We have included this information in the Figure 2 legend.

About the anti-inflammatory activity, SECO, LARI, and DCA showed inhibition toward COX-1 and COX-2. However, there’s no positive control to compare for the inhibitory effect. Furthermore, the low selectivity of COX-1/COX-2 ratio would cause some gastrointestinal side effects1. Therefore, the development and application should be considered carefully.

AUTHORS: Thank you very much for your very useful comments, we have included this information in the revised version of the manuscript.

Besides the above suggestion, there are many format mistakes in the reference part. The scientific name should be italic (references 2, 5, 8, 12, 13, 16, 17, 19-21, 24, 26, 27, 29, 30, 32, 33, 35-52, 63-65). Reference 13 lacks the journal name. The absence of volume or page should be noticed (references 32, 33, 64, 67).

AUTHORS: We have revised the references accordingly.